# Sustainable Protein-Enriched Biscuits: Effects of Mealworm Protein Powder on the Properties of Wheat Flour and Biscuit Quality

**DOI:** 10.3390/foods14173063

**Published:** 2025-08-30

**Authors:** Ao Yang, Hongrui Chen, Haixin Tian, Jianhui An, Longchen Shang, Yexing Tao, Lingli Deng

**Affiliations:** College of Biological and Food Engineering, Hubei Minzu University, Enshi 445000, China; 202430432@hbmzu.edu.cn (A.Y.); 202311982@hbmzu.edu.cn (H.C.); 202310473@hbmzu.edu.cn (H.T.); 2020049@hbmzu.edu.cn (J.A.); 2021021@hbmzu.edu.cn (L.S.); 2023060@hbmzu.edu.cn (Y.T.)

**Keywords:** wheat dough, farinograph, soda crackers, cookies, texture, PCA

## Abstract

Mealworm (*Tenebrio molitor*, LINNAEUS, 1758) is a protein-rich edible insect. In this study, low-gluten wheat flour was formulated with mealworm protein powder at various concentrations (0%, 5%, 10%, 15%, and 20%) to investigate its influence on the pasting, farinographic, and extensographic properties of low-gluten wheat flour, as well as the changes in the overall quality of the resulting biscuits (soda crackers and cookies). The viscosity of the composite flour decreased with an increasing substitution level of mealworm protein powder, and the setback significantly decreased from 69.31 ± 0.16 RVU (M0) to 19.00 ± 0.71 RVU (M20), indicating enhanced resistance to starch retrogradation. Farinographic and extensographic analyses revealed that the addition of mealworm protein powder reduced dough water absorption, significantly prolonged dough development time and stability time, and enhanced overall dough stability. However, extensibility gradually decreased, with a further reduction observed as the proofing time increased. Concurrently, the baking expansion ratio and hardness of the biscuits decreased. Specifically, for soda crackers, the baking expansion ratio decreased from 198.96 ± 3.88% (M0) to 135.74 ± 1.28% (M20), and hardness dropped from 26.40 ± 1.53 N (M0) to 6.32 ± 0.08 N (M20). For cookies, the baking expansion ratio and hardness decreased from 93.77 ± 0.72% (M0) to 86.06 ± 1.08% (M20) and from 1.76 ± 0.06 N (M0) to 1.10 ± 0.16 N (M20), respectively. The impact of mealworm protein powder (5–20%) was relatively minor in cookies but more pronounced in soda crackers, likely due to differences in formulation and processing methods. Additionally, the crunchiness of soda crackers was 3.42 times greater than that of cookies, whereas resilience was only 0.15 times that of cookies under controlled conditions. Pearson correlation analysis and principal component analysis (PCA) further elucidated the relationships between the dough properties and final product quality. Furthermore, the substitution of mealworm protein powder affected the sensory properties of the product but significantly enhanced its nutritional value, confirming the feasibility of replacing low-gluten wheat flour with mealworm protein powder and offering a theoretical foundation for its development and application in diverse biscuit formulations.

## 1. Introduction

Biscuits are essential components of the global diet [1]. They appeal to a broad consumer base because of their diversity. The global biscuit market is projected to continue its growth trajectory [2]. However, the poor nutritional value of traditional biscuits (typically high in carbohydrates and low in fiber and protein) is associated with a risk of developing metabolic or cardiovascular diseases [3]. To enhance the nutritional value of biscuits for a healthy lifestyle, fiber [4], proteins [5], or high-nutrient powders [6,7] can be added during the production process. A recently explored solution for enhancing the nutritional value of biscuits is the addition of edible insect powder [8].

Insects represent a unique food source with a history of consumption dating back approximately 7000 years [9]. However, their development has progressed relatively slowly owing to regulatory limitations and societal perceptions [10,11]. In recent years, insects have garnered increasing attention owing to their high nutritional value, safety, and palatability [12,13,14]. Edible insects are rich in proteins, carbohydrates, fats, minerals, vitamins, and other bioactive compounds [15]. Furthermore, the breeding process has a comparatively lower environmental impact and exhibits greater energy efficiency than traditional livestock farming. Research indicates that breeding edible insects not only results in high output but also has the potential to mitigate certain health risks [16]. Among the various edible insects, the mealworm is particularly notable for its high protein and fat content as well as its minimal environmental footprint [17]. In 2025, the European Commission authorized the marketing of UV-treated whole mealworm powder as a novel food ingredient that can be incorporated into various commonly consumed products such as bread and cakes [18]. It has been reported that consumers are more likely to accept eating insects when processed into powdered forms (e.g., flour) rather than consumed whole [19]. Numerous studies have successfully applied mealworm powder in various baked flour-based products. Kim et al. [20] were the first to incorporate mealworm powder into bread production. Draszanowska et al. [21] noted that incorporating mealworm powder into biscuit formulations substantially increased both the protein and fat content. However, elevated fat levels can increase the risk of oxidative rancidity, thereby compromising the product quality. Furthermore, research has indicated that the high fat content of mealworm powder may interfere with the powdering processes and transportation equipment [22]. In this context, mealworm protein powder, with a lower fat content (6.4%, ~20% less than that of conventional mealworm powder), has demonstrated certain advantages.

Low-gluten wheat flour (protein content: 7–9%; wet gluten: <25%) in biscuits can facilitate the formation of a loose and porous structure, contributing to a crispy and flaky texture, ideal for biscuits. A wide variety of biscuit types are available, and consumers typically select their preferred products based on taste and appearance [23]. To provide consumers with more robust reference information, this study was not confined to a single product category. Instead, it selected two types of biscuits with relatively high market influence—soda crackers and cookies—as the research objects.

This study investigated the effects of partially replacing low-gluten wheat flour (wet gluten < 25%) with mealworm protein powder on the rheological properties of the composite flour by evaluating its pasting, farinographic, and extensographic characteristics. Moreover, two widely consumed biscuits—soda crackers (thin and crispy) and cookies (flaky)—were formulated using the composite flour to systematically evaluate the feasibility of incorporating mealworm protein powder. The nutritional composition, color, baking expansion ratio, sensory properties, and texture of the biscuits were analyzed to evaluate the quality of the product. Pearson correlation analysis and principal component analysis (PCA) were conducted to reveal the correlations among variables.

## 2. Materials and Methods

### 2.1. Materials

Low-gluten flour (8.5% protein, 1.1% fat, 76.4% carbohydrates, 0.55% dietary fiber, 13% moisture, and 0.45% ash) was obtained from Xinxiang Xinliang Cereals Processing Co., Ltd. (Xinxiang, Henan, China). Mealworm protein powder (65% protein, 6.4% fat, 16.27% carbohydrates, 1.42% dietary fiber, 6.21% moisture, and 4.7% ash) was purchased from Qingdao Sino Crown Biological Engineering Co., Ltd. (Qingdao, China). Yeast was supplied by Angel Yeast Co., Ltd. (Yichang, China). Baking soda and white granulated sugar were purchased from Shanghai Fengwei Industrial Co., Ltd. (Shanghai, China). Common ingredients such as table salt and eggs were purchased from a local supermarket.

### 2.2. Preparation of Composite Flour Samples

The composite flour samples were prepared by substituting low-gluten wheat flour with mealworm protein powder in varying proportions, followed by thorough mixing. The substitution levels were set to 0% (M0), 5% (M5), 10% (M10), 15% (M15), and 20% (M20).

### 2.3. Pasting Properties

The pasting properties of the composite flour samples were analyzed using a Rapid Visco Analyzer (RVA-Eritm; PerkinElmer, Waltham, MA, USA). First, 25 mL of distilled water was measured, and 3 g of the composite flour was weighed. The sample was then transferred to a canister and placed in the RVA. A stirrer was inserted into the canister and moved up and down to ensure the complete dispersion of the sample. The stirrer was then attached to the connecting device and centered within the canister. In accordance with Chinese GB/T 24853-2010 [24,25] and the American Association of Cereal Chemists (AACC) method 76-21.02 (AACC, 2000) [26,27], the measurements were initiated within 1 min of sample preparation. The recorded parameters included pasting temperature, peak viscosity, peak time, trough viscosity, final viscosity, breakdown, and setback.

### 2.4. Farinographic Analysis

The farinographic properties of the dough were evaluated using a farinograph (JFZD; Beijing Dongfu Jiuheng Instrument Technology Co., Ltd., Beijing, China). In accordance with Chinese GB/T 14614-2019 [28,29] and the method AACC 54-21.01 [30,31], 300 g of composite flour was placed in a kneading bowl and preheated with stirring at 30 °C for 1 min. Subsequently, a specific amount of water was added within 25 s to hydrate the flour while kneading, thereby achieving a dough consistency of 480–520 BU. Water absorption, dough development time, stability time, and farinograph quality number were measured. Additionally, the testing duration was extended to a minimum of 12 min to assess the degree of dough softening.

### 2.5. Extensographic Analysis

An extensograph (JMLD 150; Beijing Dongfu Jiuheng Instrument Technology Co., Ltd., Beijing, China) was used to assess dough extensibility. In accordance with Chinese GB/T 14615-2019 [28,32] and the method AACC 54-10.01 [33,34], a mixture of flour, water, and 6 g NaCl was prepared under the same conditions as those employed for the farinograph test. The kneading time was controlled at 5 ± 0.1 min to ensure that the optimal dough consistency reached 500 ± 20 BU within the designated time. The dough was then removed and weighed to obtain a 150 g sample. Initially, the dough was placed in a kneading machine and passed through rollers to form cylindrical shapes. The shaped dough samples were placed in a proofing chamber and collected after three different proofing times (45, 90, and 135 min). Stretching energy, extensibility, stretching resistance, and stretch ratio were measured and analyzed.

### 2.6. Biscuit Preparation

Two types of biscuits (soda crackers and cookies) were prepared using different methods. All flour used in their production is composite flour (containing 0%, 5%, 10%, 15%, and 20% mealworm protein powder; the products were labeled M0, M5, M10, M15, and M20, respectively). The specific compositions (composite powder mass: 150 g) are listed in Table 1.

Soda crackers: Three grams of yeast was dissolved in sixty grams of milk and stirred thoroughly to ensure complete mixing. Next, 150 g of composite flour, 2 g of edible salt, and 1 g of baking soda were combined in a dough mixer (AM-CG108-1; ACA, Zhongshan, Guangdong, China) and blended evenly. Melted butter (30 g) was then added to the dry ingredients, and the dough mixer was started at a low speed while gradually incorporating the milk/yeast mixture into the flour mixture in three portions. The dough was then mixed continuously at increasing speeds until a flaky consistency was achieved. The dough was then removed and kneaded into a ball. After 30 min of fermentation at room temperature, the dough balls were rolled to a thickness of 2.5 mm using a pasta machine (YM1; Joyoung Co., Ltd., Jinan, China). The dough sheets were cut into appropriately sized pieces and perforated to prevent bulging during baking. Finally, the dough was baked in an oven (K6, Daewoo, Foshan, Guangdong, China) at 175 °C (top heat) and 155 °C (bottom heat) for 20 min. After baking, the soda crackers were cooled at room temperature for 1 h before being placed in a sealed container for storage.

Cookies: First, 100 g of butter was melted and set aside. In a separate bowl, 150 g of composite flour, 40 g of white granulated sugar, and 1 g of table salt were mixed until well combined. The dry mixture was transferred to a dough mixer. The melted butter and 30 g of egg yolks were added sequentially while continuously stirring to form a smooth dough. Once the dough was fully mixed, it was shaped into a cube using a mold and frozen for 40 min to facilitate slicing. The dough was then removed from the freezer, sliced into 1 cm thick pieces, and placed in an oven preheated to 175 °C (top heat) and 155 °C (bottom heat) for 20 min. After baking, the cookies were cooled at room temperature for 1 h before being placed in a sealed container for storage.

### 2.7. Colorimetric and Baking Expansion Ratio Analysis

A chroma meter (CS-820N, Hangzhou CHNSpec Technology Co., Ltd., Hangzhou, China) was used to evaluate the color characteristics of the biscuits. The impact of partial substitution of mealworm protein powder on the color of the final product was assessed by analyzing *L** (lightness/darkness), *a** (redness/greenness), and *b** (blueness/yellowness). Each sample was measured three times, and the average value was calculated to ensure accuracy.

The baking expansion ratios of the biscuits were determined according to the method described by Xie et al. [35]. A digital Vernier caliper (Evite, Henan Bangte Measuring Tools Co., Ltd., Shangqiu, Henan, China) was used to measure the thickness of the dough sheets before baking and the thickness of the biscuits after baking. Each biscuit sample was appropriately labeled to ensure a one-to-one correspondence between the dough and its baked counterpart. The baking expansion ratio was calculated using the following formula:$$X\left(\%\right)\;=\;\frac{h_{2}}{h_{1}}\;\times \;100\%$$
where *X* represents the baking expansion ratio, *h*_1_ is the thickness of the dough sheet before baking (mm), and *h*_2_ is the thickness of the biscuit after baking (mm). Each measurement was repeated thrice.

### 2.8. Texture Profile Analysis of Biscuit

According to the method described by Yang et al. [36], a texture analyzer (TA-XT Plus, Stable Micro Systems Ltd., Surrey, UK) equipped with a 3-point bending probe (TA/3PB) was used to perform a 3-point break test on the biscuits in the single test mode. The bending device was calibrated at a height of 10 mm above the sample surface. The pre-test, test, and post-test speeds were set to 1 mm/s, with a trigger force of 5 g and a target displacement of 5 mm. The hardness, crunchiness, and resilience of the biscuits were measured. In this context, the crunchiness refers to the distance from the point at which the probe touches the biscuit until it breaks. A higher crunchiness value indicated a less crispy biscuit. Each sample was measured three times, and the average was calculated.

### 2.9. Proximate Compositions

Mealworm protein powder, low-gluten wheat flour, and the biscuits were sent to the Pony Testing Group (Beijing, China) for proximate composition analysis. The protein content was determined in accordance with the Chinese GB 5009.5-2016 standard [37], while the fat content was assessed following the guidelines of Chinese GB 5009.6-2016 [38,39]. The carbohydrate content was analyzed according to Chinese GB 28050-2011 [27,40]. In addition, the moisture and ash determinations adhered to the standards outlined in Chinese GB 5009.3-2016 [41] and Chinese GB 5009.4-2016 [35,42], respectively.

### 2.10. Sensory Evaluation of Biscuits

The sensory quality of biscuits is an important factor that influences the overall quality of a product and is influenced by human vision, smell, taste, hearing, and touch. The experiment was conducted by 40 evaluators (20 males and 20 females; age group: 20–30 years old). The biscuit samples were packaged in the same manner and randomly assigned three-digit codes. A 9-point hedonic scale was used to assess the appearance, aroma, texture, taste, saltiness, and general acceptability of the samples. Specifically, a score of 9 corresponded to “extremely like,” whereas a score of 1 indicated “extremely dislike.”

### 2.11. Statistical Analysis

All experiments were conducted in triplicate. The experimental data were organized using Microsoft Excel 2016. The significance and correlations among the variables were analyzed using OriginPro 2024b (OriginLab, Northampton, MA, USA), with statistical significance defined as *p* < 0.05. Data were presented as mean ± standard deviation. Pearson correlation analysis was performed to evaluate the relationships between dough properties and biscuit quality, with significance set at *p* < 0.05 and high significance at *p* < 0.01. Additionally, PCA was employed to explore the relationship between the mealworm protein powder substitution levels and the quality parameters of the two biscuit types.

## 3. Results and Discussion

### 3.1. Pasting Properties of Flour

Pasting characteristics reflect the rheological behavior of gelatinized starch and are closely related to product quality [43]. The results of the composite flour analyses are presented in Figure 1 and Table 2. The incorporation of mealworm protein powder significantly decreased the peak viscosity, trough viscosity, final viscosity, and setback of the low-gluten flour. This reduction in viscosity may be attributed to the increased protein content and decreased starch content resulting from the substitution. Consequently, the gluten network was modified, promoting the encapsulation of starch granules within the gluten matrix [44]. Setback is closely associated with the retrogradation and aging behaviors of starch. A lower setback suggests that mealworm protein powder reduces the stability of starch granules, inhibits the recrystallization of starch molecules, and consequently delays starch aging [45]. Breakdown (defined as the difference between peak viscosity and trough viscosity) exhibited an increasing trend following the addition of mealworm protein powder compared to M0, indicating that substitution altered the thermal stability of the composite flour system and reduced its heat resistance and tolerance during gelatinization [46]. However, the breakdown of M20 was markedly reduced, suggesting that a high substitution level of mealworm protein powder may enhance the resistance of starch paste to shear thinning [47]. The pasting temperature refers to the minimum temperature required for starch gelatinization and is influenced primarily by the starch content and its compositional characteristics [48]. Mealworm protein powder did not significantly affect the pasting temperatures at lower substitution levels. In contrast, at 20% substitution, the pasting temperature of the composite flour increased significantly, reaching 86.74 ± 0.63 °C. This increase may be attributed to the relatively high resistance to swelling and rupture associated with high concentrations of mealworm protein powder, which influences heat-transfer dynamics [49]. A similar trend in pasting temperature was observed by Xie et al., who replaced whole wheat flour with mealworm powder at substitution levels ranging from 0 to 20% in steamed bread formulations [27]. Additionally, no significant variation was observed in the peak time.

### 3.2. Farinographic Properties of Dough

Farinographic properties reflect the resistance of flour to deformation during kneading and serve as key indicators for evaluating the processing performance of flour. The farinographic profiles of the low-gluten dough prepared with varying levels of mealworm protein powder—control (0%) and substitution levels of 5%, 10%, 15%, and 20%—are presented in Figure 2. The overall change in the curve indicates that the dough underwent the entire process of formation, stabilization, and attenuation. The composite flour samples of M0 and M5 showed a rapid increase in the initial ascending stage of the curve, suggesting that the gluten network formed quickly. At higher substitution levels (>5%), the curve showed a relatively gentle rise followed by a somewhat more rapid increase, indicating that gluten formation was relatively slow. The decrease in the speed of gluten formation may be attributed to the interaction between the mealworm protein powder and low-gluten flour, which delays gluten hydration and development [50]. After reaching its peak, the curve entered a stable period. Compared to the control group, the addition of mealworm protein powder significantly prolonged the stable period of the dough. Subsequently, the curve showed a downward trend, suggesting that the dough had entered the attenuation period. The starch/gluten network structure of the dough became unstable and began to decompose as the mixing time increased, leading to a gradual reduction in the overall strength and stability of the dough [51]. The curve decreased more significantly at M5, indicating that at this concentration, mealworm protein powder disrupted the three-dimensional structure of gluten protein and had poor elasticity [52].

The corresponding farinographic parameters are listed in Table 3. Water absorption is closely associated with the hydration capacity of gluten proteins and plays a critical role in determining product quality and safety [53,54]. The incorporation of flour rich in hydrophilic proteins into dough typically enhances dough’s water absorption through the increase in the number of water-binding sites. However, mealworm protein powder, a starch-free protein source, exhibited the opposite effect, despite its relatively high protein content. A decrease in dough water absorption was observed when the incorporation of mealworm protein powder exceeded 5%. This phenomenon is speculated to be related to the amino acid composition of mealworm protein powder. Differences in the hydrophilic/hydrophobic properties of different amino acid residues may change the overall water-binding capacity of a protein. Gonzalez et al. [55] reached a similar conclusion. In addition, the “dilution effect” of mealworm protein powder may contribute to this phenomenon. As the concentration of mealworm protein powder increased, the relative gluten protein content of the composite flour gradually decreased. Given that gluten proteins are the core component determining dough water absorption, a reduction in gluten content directly diminishes the number of available water-binding sites and the pore capacity of the dough’s protein network, ultimately manifested as a reduction in the overall water absorption rate [27,56].

With increasing substitution levels of mealworm protein powder, the dough development time was significantly prolonged, rising from 1.27 ± 0.12 min (M0) to 8.10 ± 0.60 min (M20). This extension may be attributed to the effect of mealworm protein powder on the hydration of proteins. During mixing, gluten proteins absorb water and swell; however, the arrangement and depolymerization of these proteins may be disrupted [57,58]. Additionally, the dough stability time increased from 1.17 ± 0.06 min (M0) to 8.57 ± 0.29 min (M20), suggesting that the incorporation of mealworm protein powder facilitated the formation of a more compact and elastic gluten network, thereby enhancing the structural stability of the dough [59]. Jia et al. [60] reported that the dough development time, stability time, and farinograph quality number are generally positively correlated with dough strength. The observed increase in the farinograph quality number following the addition of mealworm protein powder further supports this finding [60]. Moreover, the degree of softening of the dough decreased at higher substitution levels (>5%), suggesting that a higher concentration of mealworm protein powder enhanced the dough’s toughness and resistance to mechanical damage. However, the degree of softening increased and then decreased at M15, probably due to the destruction of the gluten network by mealworm protein powder fibers at this concentration [61]. Cappelli et al. found the same rule when they replaced wheat flour with chickpea flour at a substitution level of 15%, with the degree of softening of the flour increasing to 48.33 UB [50].

### 3.3. Extensographic Properties of Dough

Extensographic analysis provides information on the viscoelastic behavior of flour [62]. The effects of different substitution levels of mealworm protein powder and proofing times on the extensographic properties are summarized in Figure 3. The incorporation of mealworm protein powder significantly decreased both the stretching energy and extensibility of the dough under the same proofing time. Extensibility reflects the capacity of the dough to stretch under tensile stress and serves as an indicator of the flexibility of the gluten network. With increasing substitution levels of mealworm protein powder, dough extensibility under proofing conditions of 45, 90, and 135 min decreased from 116.00 ± 1.00 mm (M0) to 73.33 ± 0.58 mm (M20), from 100.00 ± 4.36 mm (M0) to 70.00 ± 2.00 mm (M20), and from 109.67 ± 7.23 mm (M0) to 63.00 ± 1.00 mm (M20), respectively, perhaps because the relative coarseness of the mealworm protein powder leads to an increase in friction with the protein network fiber during the stretching process, thus accelerating the fracture of the gluten network [63]. These findings are consistent with those reported by Bresciani et al. [64], who observed a significant decrease in dough extensibility following the substitution of wheat flour with cricket flour. Moreover, the stretch ratio (calculated as stretching resistance over extensibility) of the dough increased significantly compared to that of M0; this positive correlation with the substitution level of mealworm protein powder suggests that higher concentrations of mealworm protein powder promote protein aggregation. As a result, interactions between starch granules and mealworm proteins during extension contributed to a denser and more compact dough texture [65,66].

Balanced stretching resistance is essential for producing high-quality baked goods [67]. The stretching resistance of M20 increased significantly after 45 min, indicating an increase in dough hardness [68]. Furthermore, the incorporation of mealworm protein powder did not significantly affect the dough stretching resistance at proofing times of 45 and 90 min. However, when the proofing time was extended to 135 min, a notable increase in stretching resistance was observed, suggesting that the proofing duration influenced the dough’s extensographic properties. As the proofing time increased, the stretching energy initially increased and then decreased, whereas both the stretching resistance and stretch ratio generally exhibited an upward trend. These results indicate that dough strength improves with prolonged proofing. However, excessive proofing may negatively influence dough extensibility and ultimately compromise product quality.

### 3.4. Analysis of Color and Baking Expansion Ratio of Biscuits

The visual appearances of the resulting soda crackers and cookies are shown in Figure 4. Progressive darkening of the biscuit color was observed with increasing mealworm protein powder substitution levels, along with surface textural changes characterized by increased porosity and roughness. The color parameters (L*: brightness/darkness; a*: redness/greenness; b*: yellowness/blueness) and baking expansion ratios are detailed in Table 4 and Table 5. The chromatic properties of the raw materials have been shown to influence product pigmentation directly, which is a critical determinant of consumer acceptability [69,70]. A reduction in the L* values was observed in both soda crackers and cookies with increasing mealworm protein powder content, showing near-linear trends from 75.65 ± 0.82 (M0) to 50.87 ± 0.96 (M20) for soda crackers and from 71.75 ± 0.15 (M0) to 50.88 ± 0.73 (M20) for cookies. The addition of mealworm protein powder led to a significant increase in both a* and b* values in the biscuits in relation to those observed in the control group. This chromatic shift is likely attributable to the intrinsic yellowish-brown pigmentation of the mealworm protein powder and the potential enhancement of the Maillard reaction resulting from the added proteins and free amino acids [71]. Notably, minimal variation in the b* values was recorded in cookies when the substitution level increased from 5% to 20%, suggesting that the concentration of mealworm protein powder had no substantial impact on b* in cookie formulations. However, a* also showed no significant change at higher substitution levels (>5%), which is considerably different from the trend observed for soda crackers. This contrast in trends may be because the high butter content of cookies can neutralize the color effect caused by the different concentrations of mealworm protein powder.

The incorporation of mealworm protein powder significantly influenced the baking expansion ratio of soda crackers and cookies. Overall, the baking expansion ratio of soda crackers and cookies decreased with the addition of mealworm protein powder, reaching their lowest values at M20, which were 135.74 ± 1.28% and 86.06 ± 1.08%, respectively. The decrease in expansion suggests that the ability of the composite dough to retain gas during baking was impaired [35]. Notably, the baking expansion ratio of cookies was not only significantly lower than that of soda crackers but also fell below 100%. This phenomenon may be attributed to the high fat and sugar content affecting the gluten network structure and weakening the gas retention capacity [72]. The growth and recrystallization of ice crystals during the freezing stage cause volume expansion during cookie preparation. Upon baking, the subsequent melting of the ice crystals leads to a reduction in the product volume owing to the high-temperature conditions, indicating that the gluten network structure was affected during the freeze/thaw process and the redistribution of water in the dough [73].

### 3.5. Texture Analysis of Biscuits

The textural characteristics of products are well established as critical determinants of consumer acceptance [74]. The specific texture parameters of the soda crackers and cookies are shown in Figure 5. The addition of mealworm protein powder generally reduced the hardness of biscuits although the effect was relatively weaker in cookies at the M5 substitution level. In contrast, mealworm protein powder demonstrated a concentration-dependent effect on the hardness of soda crackers, with values decreasing from 26.40 ± 1.53 N (M0) to 6.32 ± 0.08 N (M20) as the substitution level increased. This effect is possibly due to the substitution of mealworm protein powder, which introduced additional fat, thus inhibiting the evaporation of moisture during the baking process [21,55]. Xie et al. [35] reported a reduction in biscuit hardness following the incorporation of mealworm powder. Furthermore, compared to the biscuits in the control group, the addition of mealworm protein powder had no significant effect on the crunchiness of soda crackers, whereas the crunchiness of M20 cookies was significantly reduced, indicating that high-concentration mealworm protein supplementation negatively affected texture perception. The resilience (i.e., the rubbery state) of biscuits increased with the addition of mealworm protein powder, which increased the number of covalent bonds and protein aggregation. Mealworm proteins can form a soft protein gel structure with a certain degree of elasticity, enhancing their rebound resilience [53,75]. However, cookies are characterized by a high lipid content. The incorporation of mealworm protein powder leads to an uneven distribution of lipids, resulting in the encapsulation of starch granules and subsequent interference with gluten network formation [35].

### 3.6. Multivariate Statistical Analysis

Pearson correlation analysis was conducted under standardized experimental conditions to examine the relationship between dough properties and biscuit quality characteristics at different substitution levels of mealworm protein powder (Figure 6). The viscosity parameters of the composite flour were significantly positively correlated with the L* value and hardness of both types of biscuits and significantly negatively correlated with the a* value. In contrast, no statistically significant correlation was observed between breakdown and any biscuit quality attribute. In addition, dough extensibility showed a strong positive correlation with biscuit hardness, indicating that both extensibility and hardness decreased as the substitution level of mealworm protein powder increased. Water absorption was significantly positively correlated with the hardness and baking expansion ratio of soda crackers, as well as with the hardness and crunchiness of cookies, underscoring moisture content as a key determinant of the final product quality. Notably, the crunchiness of soda crackers did not exhibit a significant correlation with any of the measured dough properties. However, the crunchiness of the cookies was significantly positively correlated with peak viscosity, trough viscosity, and water absorption, despite being derived from the same composite powder. The different responses of soda crackers and cookies is attributed to the fact that different auxiliary materials were added to the two types of biscuits, resulting in significant differences in the chemical composition, which led to certain differences in the correlations between the dough characteristics and product quality.

PCA was employed to investigate the relationships among the quality parameters of soda crackers and cookies at various substitution levels of mealworm protein powder. A PCA biplot (Figure 7) was generated to visualize the correlation structure among the measured variables based on the evaluation of color attributes (*L**, *a**, *b**), baking expansion ratio, hardness, crunchiness, and resilience. The first principal component (PC1, 54.6%) was influenced primarily by the baking expansion ratio, hardness, crunchiness, and resilience, with a negative loading for resilience, indicating that samples with lower PC1 scores exhibited higher resilience. From the perspective of biscuit quality characteristics, the addition of mealworm protein powder reduced the baking expansion ratio, hardness, and crunchiness, while increasing the resilience of biscuits, which also confirmed this finding. Color parameters constituted the second major factor (PC2, 35.8%), showing positive correlations with *a** and *b**, and a negative correlation with *L**. Notably, resilience was negatively associated with crunchiness, baking expansion ratio, and hardness, whereas hardness was positively correlated with the baking expansion ratio. This association may be explained by the weakening of the gluten network, which reduces biscuit hardness and limits the gas retention capacity of the dough, thereby lowering the baking expansion ratio [46]. Overall, the sample points of both biscuit types shifted along the horizontal and vertical axes of the PCA plot as the substitution level of mealworm protein powder increased, indicating that the powder concentration influenced the biscuit quality. In addition, the sample points corresponding to soda crackers were located entirely on the right side of the PC1 axis, whereas those for cookies were located on the left and clustered near the resilience vector. This distribution suggests that cookies exhibit higher resilience, whereas soda crackers show greater hardness, crunchiness, and baking expansion ratio. These differences highlight the close relationship between the product quality characteristics, distinct manufacturing processes, and ingredient compositions used for each biscuit type.

### 3.7. Nutritional Values

The nutritional component ratios of mealworm protein powder, low-gluten wheat flour, and the corresponding biscuit products are shown in Figure 8. Notably, mealworm protein powder exhibited an exceptionally high protein content, reaching up to 65%, and the fat content was significantly higher than that of low-gluten wheat flour. As the substitution level of mealworm protein powder increased, the protein, fat, carbohydrate, moisture, and ash contents of the biscuits underwent corresponding changes. Specifically, the protein content of soda crackers and cookies increased from 9.05% to 18.4% and from 7.61% to 10.06%, respectively, with increasing substitution levels. In addition, significant differences were observed in the proximate compositions of the two types of biscuits. The fat content of the cookies was nearly twice that of soda crackers, which could be attributed to differences in raw material proportions, particularly the higher amount of butter added during cookie preparation.

### 3.8. Sensory Evaluation of Biscuits

As illustrated in Figure 9, the incorporation of the mealworm protein powder significantly affected the sensory attributes of the biscuits. In general, the scores for appearance, aroma, texture, taste, saltiness, and overall acceptability of both types of biscuits decreased with an increase in mealworm protein powder substitution levels. Among these attributes, the score variations for aroma and overall acceptability were the most pronounced, indicating that the odor of the mealworm protein powder was the primary factor influencing consumer acceptance, exerting a negative effect on product quality. This adverse impact may also be related to psychological factors, as mealworms are often associated with negative perceptions such as disgust, strangeness, or curiosity [76]. Furthermore, notable differences were observed in the sensory scores between the two biscuit types; the overall acceptability of cookies was higher than that of soda crackers. This observation suggests that high-fat and high-sugar formulations can partially mask the negative sensory effects induced by mealworm protein powder. Overall, when the substitution level of the mealworm protein powder was set to M5, the sensory scores differed only slightly or were even higher than those of M0, indicating that the product at this substitution level retained both palatable and nutritious characteristics.

## 4. Conclusions

Based on the analysis of the research findings, mealworm protein powder shows considerable potential as an alternative protein source for biscuit production, given that its protein content is 56.5% higher per 100 g than that of low-gluten wheat flour. In terms of dough properties, increasing the substitution level of mealworm protein powder reduced the viscosity of gelatinized starch, thereby exerting a negative influence on the pasting behavior. Additionally, the incorporation of mealworm protein powder decreased the water absorption and extensibility of dough, while simultaneously increasing the farinograph quality number and stretch ratio, which are parameters indicative of enhanced dough strength. Elevated inclusion levels also affected the biscuit quality by reducing the hardness and baking expansion ratio. Overall, the cookies had relatively low hardness and showed low dependence on the concentration of mealworm protein powder. These observations were further supported by multivariate statistical analysis. A higher sensory score indicated that the cookies better met the needs of consumers than soda crackers with mealworm protein powder did. Considering the significant impact of mealworm protein powder on biscuit sensory score and the minimal differences observed at lower substitution levels, it is recommended that the substitution level should not exceed 5%. At this level, no significant difference in hardness was observed compared to the control group. However, the odor and color of mealworm protein powder remain critical factors influencing consumer acceptance. Therefore, future studies should prioritize pretreatment methods aimed at mitigating these attributes and extending the shelf life to improve market potential and consumer appeal.

## Figures and Tables

**Figure 1 foods-14-03063-f001:**
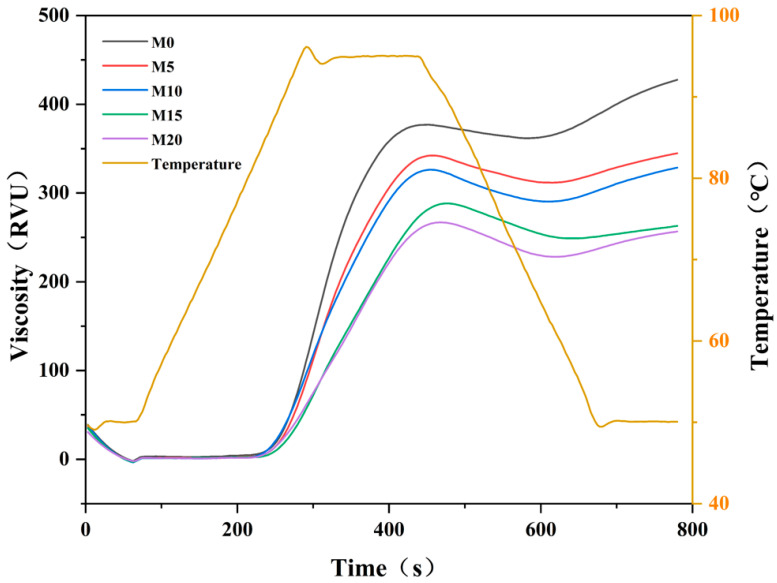
RVA curves of low-gluten wheat flour formulated with 0% (M0), 5% (M5), 10% (M10), 15% (M15), and 20% (M20) mealworm protein powder.

**Figure 2 foods-14-03063-f002:**
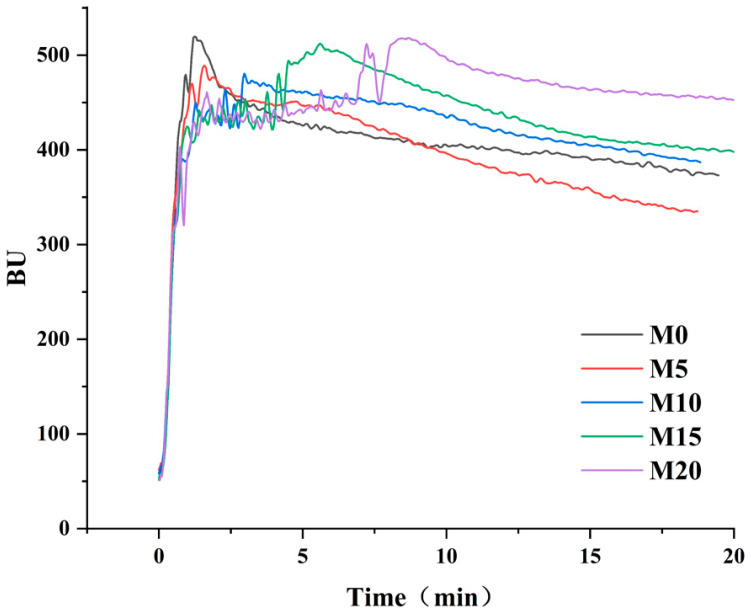
Farinographic curves of low-gluten wheat flour formulated with 0% (M0), 5% (M5), 10% (M10), 15% (M15), and 20% (M20) mealworm protein powder.

**Figure 3 foods-14-03063-f003:**
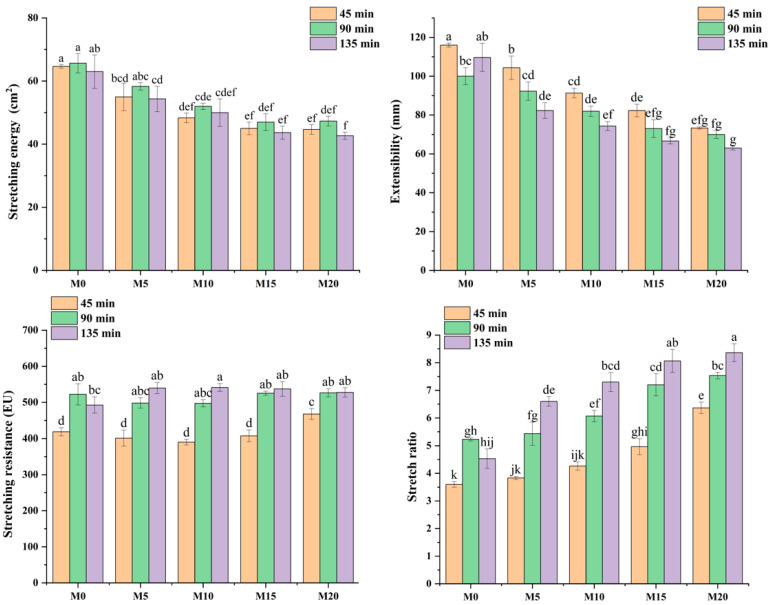
Extensographic properties of low-gluten wheat flour formulated with 0% (M0), 5% (M5), 10% (M10), 15% (M15), and 20% (M20) mealworm protein powder. Different letters indicate significant differences (*p* < 0.05).

**Figure 4 foods-14-03063-f004:**
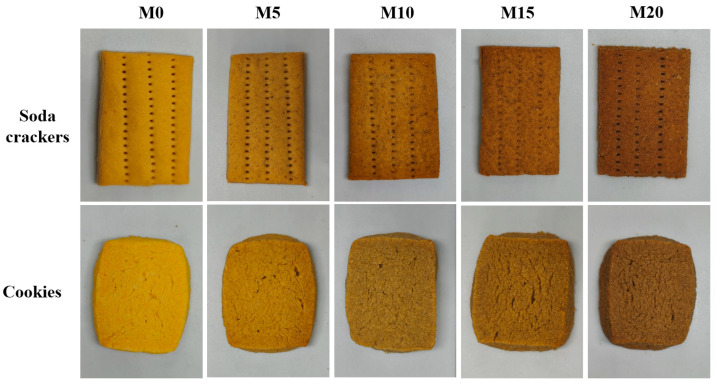
Visual appearance of soda crackers and cookies prepared with 0% (M0), 5% (M5), 10% (M10), 15% (M15), and 20% (M20) mealworm protein powder.

**Figure 5 foods-14-03063-f005:**
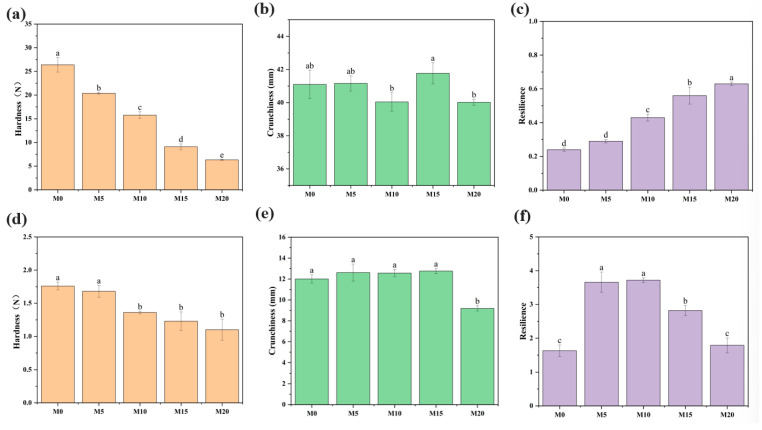
Effects of substitution levels of mealworm protein powder on the texture parameters of biscuits. Soda crackers (**a**–**c**); cookies (**d**–**f**). Different letters above the columns indicate statistically significant differences (*p* < 0.05). Products containing 0%, 5%, 10%, 15%, and 20% mealworm protein powder are labeled M0, M5, M10, M15, and M20, respectively.

**Figure 6 foods-14-03063-f006:**
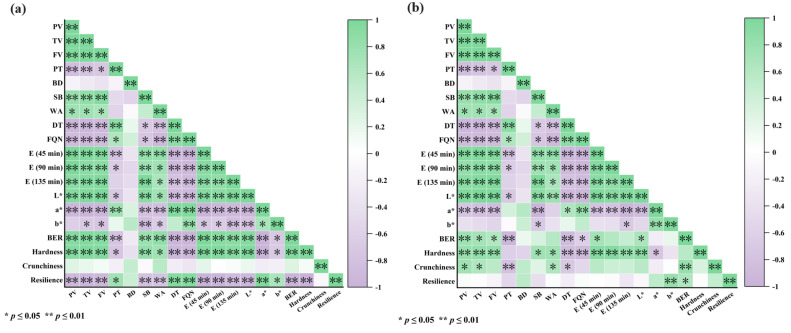
Pearson correlation analysis between dough properties and biscuit quality at various mealworm protein powder substitution levels: (**a**) soda crackers; (**b**) cookies. Abbreviations: PV, peak viscosity; TV, trough viscosity; FV, final viscosity; PT, pasting temperature; BD, breakdown; SB, setback; WA, water absorption; DT, dough development time; FQN, farinograph quality number; E, extensibility; *L**, brightness/darkness; *a**, redness/greenness; *b**, yellowness/blueness; BER, baking expansion ratio.

**Figure 7 foods-14-03063-f007:**
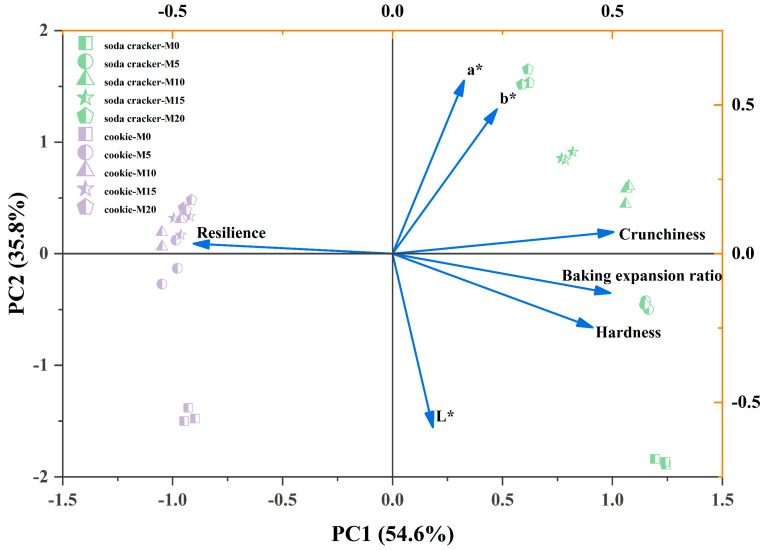
Principal component analysis (PCA) of the physical parameters associated with varying substitution levels of mealworm protein powder in soda crackers and cookies. Products containing 0%, 5%, 10%, 15%, and 20% mealworm protein powder are labeled M0, M5, M10, M15, and M20, respectively.

**Figure 8 foods-14-03063-f008:**
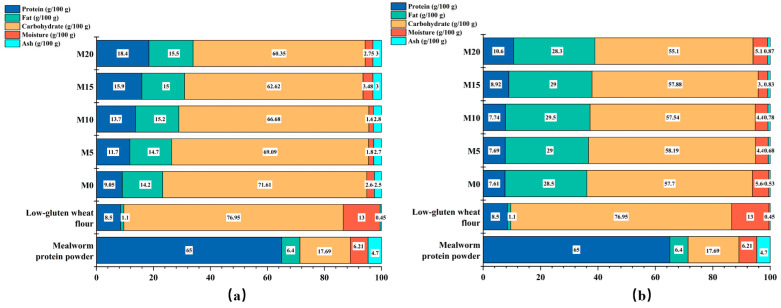
Proximate composition of the mealworm protein powder, low-gluten wheat flour, and soda crackers (**a**) and cookies (**b**) with various mealworm protein powder substitution levels. Products containing 0%, 5%, 10%, 15%, and 20% mealworm protein powder are labeled M0, M5, M10, M15, and M20, respectively.

**Figure 9 foods-14-03063-f009:**
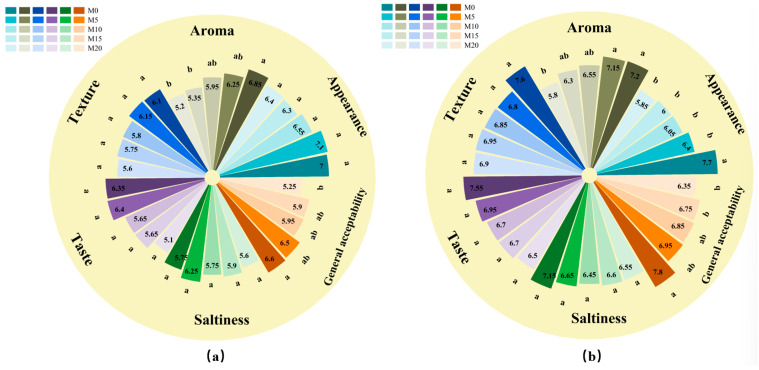
Sensory scores of soda crackers (**a**) and cookies (**b**) with various mealworm protein powder substitution levels. Different superscript letters within the same color group indicate statistically significant differences (*p* < 0.05). Products containing 0%, 5%, 10%, 15%, and 20% mealworm protein powder were labeled M0, M5, M10, M15, and M20, respectively.

**Table 1 foods-14-03063-t001:** Formulations for soda crackers and cookies (based on composite flour weight, 100%).

	Flour (%)	Mealworm Protein Powder (%)	Yeast (%)	Table Salt (%)	White Granulated Sugar (%)	Milk (%)	Butter (%)	Egg Yolk (%)	Baking Soda (%)
Soda crackers									
M0	100	-	2	1.33	-	40	20	-	0.67
M5	95	5	2	1.33	-	40	20	-	0.67
M10	90	10	2	1.33	-	40	20	-	0.67
M15	85	15	2	1.33	-	40	20	-	0.67
M20	80	20	2	1.33	-	40	20	-	0.67
Cookies									
M0	100	-	-	0.67	26.67	-	66.67	20	-
M5	95	5	-	0.67	26.67	-	66.67	20	-
M10	90	10	-	0.67	26.67	-	66.67	20	-
M15	85	15	-	0.67	26.67	-	66.67	20	-
M20	80	20	-	0.67	26.67	-	66.67	20	-

**Table 2 foods-14-03063-t002:** Pasting characteristics of the composite flour of low-gluten wheat flour and mealworm protein powder.

	Peak Viscosity (RVU)	Trough Viscosity (RVU)	Breakdown (RVU)	Final Viscosity (RVU)	Setback (RVU)	Peak Time (min)	Pasting Temperature (°C)
M0	376.46 ± 1.20 ^a^	359.95 ± 0.56 ^a^	16.51 ± 0.64 ^d^	429.25 ± 0.72 ^a^	69.31 ± 0.16 ^a^	7.20 ± 0.00 ^a^	80.63 ± 1.34 ^b^
M5	329.68 ± 9.01 ^b^	308.35 ± 7.92 ^b^	21.33 ± 1.09 ^c^	336.84 ± 12.67 ^b^	28.49 ± 4.75 ^bc^	7.15 ± 0.09 ^a^	81.19 ± 1.07 ^b^
M10	323.79 ± 0.04 ^b^	296.21 ± 1.36 ^b^	27.58 ± 1.40 ^b^	326.58 ± 3.36 ^b^	30.37 ± 1.99 ^b^	7.20 ± 0.00 ^a^	80.22 ± 0.47 ^b^
M15	293.80 ± 0.14 ^c^	259.31 ± 0.14 ^c^	34.49 ± 0.00 ^a^	274.69 ± 3.37 ^c^	15.38 ± 3.23 ^d^	7.20 ± 0.00 ^a^	82.57 ± 2.16 ^b^
M20	237.97 ± 8.33 ^d^	220.08 ± 9.89 ^d^	17.89 ± 1.56 ^cd^	239.08 ± 10.60 ^d^	19.00 ± 0.71 ^cd^	7.20 ± 0.00 ^a^	86.74 ± 0.63 ^a^

Values are expressed as mean ± standard deviation (*n* = 3). Different superscript letters within the same column indicate statistically significant differences (*p* < 0.05). The flour was substituted by mealworm protein powder at weight ratios of 0% (M0), 5% (M5), 10% (M10), 15% (M15), and 20% (M20), respectively.

**Table 3 foods-14-03063-t003:** Farinographic properties of composite dough prepared from low-gluten wheat flour and mealworm protein powder.

	Water Absorption (%)	Dough Development Time (min)	Stability Time (min)	Degree of Softening (FU)	Farinograph Quality Number (mm)
M0	57.20 ± 0.10 ^a^	1.27 ± 0.12 ^d^	1.17 ± 0.06 ^d^	114.67 ± 7.64 ^b^	18.33 ± 0.58 ^d^
M5	57.43 ± 0.23 ^a^	1.70 ± 0.10 ^d^	3.73 ± 0.47 ^c^	137.00 ± 11.27 ^a^	29.33 ± 3.51 ^c^
M10	55.33 ± 0.06 ^c^	3.30 ± 0.26 ^c^	7.00 ± 0.61 ^b^	77.67 ± 5.69 ^c^	81.67 ± 5.86 ^b^
M15	56.53 ± 0.21 ^b^	5.73 ± 0.15 ^b^	6.10 ± 0.70 ^b^	108.33 ± 1.15 ^b^	79.00 ± 1.73 ^b^
M20	54.63 ± 0.06 ^d^	8.10 ± 0.60 ^a^	8.57 ± 0.29 ^a^	67.33 ± 2.08 ^c^	101.67 ± 5.59 ^a^

Values are expressed as mean ± standard deviation (*n* = 3). Different superscript letters within the same column indicate statistically significant differences (*p* < 0.05). The flour was substituted by mealworm protein powder at weight ratios of 0% (M0), 5% (M5), 10% (M10), 15% (M15), and 20% (M20), respectively.

**Table 4 foods-14-03063-t004:** Effects of substitution levels of mealworm protein powder on the color parameters and baking expansion ratios of soda crackers.

	*L** (Brightness/Darkness)	*a** (Redness/Greenness)	*b** (Yellowness/Blueness)	Baking Expansion Ratio (%)
M0	75.65 ± 0.82 ^a^	4.25 ± 0.05 ^e^	27.36 ± 0.30 ^d^	198.96 ± 3.88 ^a^
M5	64.48 ± 0.10 ^b^	6.58 ± 0.15 ^d^	30.70 ± 0.20 ^c^	183.23 ± 2.54 ^b^
M10	60.76 ± 0.47 ^c^	9.86 ± 0.58 ^c^	33.12 ± 0.20 ^a^	162.83 ± 2.76 ^c^
M15	56.39 ± 1.11 ^d^	11.80 ± 0.29 ^b^	31.65 ± 0.18 ^b^	156.08 ± 0.92 ^c^
M20	50.87 ± 0.96 ^e^	14.84 ± 0.12 ^a^	31.87 ± 0.35 ^b^	135.74 ± 1.28 ^d^

Values are expressed as mean ± standard deviation (*n* = 3). Different superscript letters within the same column indicate statistically significant differences (*p* < 0.05). The flour was substituted by mealworm protein powder at weight ratios of 0% (M0), 5% (M5), 10% (M10), 15% (M15), and 20% (M20), respectively.

**Table 5 foods-14-03063-t005:** Effects of substitution levels of mealworm protein powder on the color parameters and baking expansion ratios of cookies.

	*L** (Brightness/Darkness)	*a** (Redness/Greenness)	*b** (Yellowness/Blueness)	Baking Expansion Ratio (%)
M0	71.75 ± 0.15 ^a^	2.25 ± 0.13 ^c^	26.48 ± 0.38 ^b^	93.77 ± 0.72 ^a^
M5	64.11 ± 0.47 ^b^	6.60 ± 0.38 ^b^	29.87 ± 0.93 ^a^	94.07 ± 3.04 ^a^
M10	60.49 ± 0.97 ^c^	7.60 ± 0.09 ^a^	29.89 ± 0.75 ^a^	94.14 ± 2.71 ^a^
M15	55.81 ± 0.92 ^d^	7.59 ± 0.30 ^a^	29.05 ± 0.63 ^a^	91.23 ± 0.66 ^ab^
M20	50.88 ± 0.73 ^e^	7.72 ± 0.15 ^a^	28.56 ± 0.63 ^a^	86.06 ± 1.08 ^b^

Values are expressed as mean ± standard deviation (*n* = 3). Different superscript letters within the same column indicate statistically significant differences (*p* < 0.05). The flour was substituted by mealworm protein powder at weight ratios of 0% (M0), 5% (M5), 10% (M10), 15% (M15), and 20% (M20), respectively.

## Data Availability

The original contributions presented in the study are included in the article, further inquiries can be directed to the corresponding author.

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
