# Peer review of "Sustainable Protein-Enriched Biscuits: Effects of Mealworm Protein Powder on the Properties of Wheat Flour and Biscuit Quality"

_foods, 2025, doi:10.3390/foods14173063_

Round 1
Reviewer 1 Report (Previous Reviewer 1)
Comments and Suggestions for Authors
Introduction, lines 44-49: Although there is................. Instead of writing this, please mention the names of nutrients that are deficient in biscuits and why we need to add additional sources to enrich them.
Line no 71-81, rewrite as
This study investigated the effect of partially replacing low-gluten wheat flour (wet gluten <25%) with mealworm protein powder on the rheological properties of the composite flour by evaluating its pasting, farinographic, and extensographic characteristics. Moreover, two widely consumed biscuits -Soda crackers (thin and crispy) and cookies (flaky)—were formulated using the composite flour to systematically evaluate the feasibility of incorporating mealworm protein powder. The nutritional composition, color, baking expansion ratio, sensory properties, and texture of the biscuits were analyzed to evaluate the quality of the product. Pearson correlation analysis and principal component analysis (PCA) were used to reveal the correlations among variables.
Line no 256, Rewrite as
The composite flour samples of M0 and M5 showed a rapid increase in the initial ascending stage of the curve, suggesting that the gluten network formed quickly.
During reviewing, I have found a similar paper entitled "Effect of Partial Substitution of Flour with Mealworm(Tenebrio molitor L.) Powder on Dough and Biscuit Properties by Xinyuan Xie, Zhihe Yuan, Kai Fu, Jianhui An, and Lingli Deng * July 2022 Foods 11(14):2156. DOI: 10.3390/foods11142156 has been published. How is your previous paper different from the current manuscript?

English should be improved.
Author Response
Comments:
1.Introduction, lines 44-49: Although there is.................Instead of writing this, please mention the names of nutrients that are deficient in biscuits and why we need to add additional sources to enrich them.
Response: Thank you for your careful suggestion. We have made the necessary modification to the corresponding positions, and as indicated in lines 42-47. However, the poor nutritional value of traditional biscuits is associated with the risk of developing metabolic or cardiovascular diseases. To enhance the nutritional value of biscuits for a healthy lifestyle, fibers, proteins, or high-nutrition powders are usually added during the production process. A recently explored solution to enhancing the nutritional value of biscuits is the addition of edible insect powders. We have added references to support it.
2.Line no 71-81, rewrite as
This study investigated the effect of partially replacing low-gluten wheat flour (wet gluten <25%) with mealworm protein powder on the rheological properties of the composite flour by evaluating its pasting, farinographic, and extensographic characteristics. Moreover, two widely consumed biscuits -Soda crackers (thin and crispy) and cookies (flaky)—were formulated using the composite flour to systematically evaluate the feasibility of incorporating mealworm protein powder. The nutritional composition, color, baking expansion ratio, sensory properties, and texture of the biscuits were analyzed to evaluate the quality of the product. Pearson correlation analysis and principal component analysis (PCA) were used to reveal the correlations among variables.
Response: Thank you for providing such detailed guidance on crafting this section. It has been incredibly helpful for refining my writing skills. We have reviewed this part of the content carefully, and compared to the previous version, the language is now much more concise. By unifying the descriptions of the products’ quality characteristics and clarifying the significance of studying these two types of biscuits in this context, the research logic and value of the paper are more prominently highlighted. This not only strengthens the academic rigor of the work but also helps readers quickly grasp the overall framework of the text. We have also made corresponding modifications to the content of lines 75-78 to avoid repetition.
3.Line no 256, Rewrite as
The composite flour samples of M0 and M5 showed a rapid increase in the initial ascending stage of the curve, suggesting that the gluten network formed quickly.
Response: Thank you for providing such detailed guidance on crafting this section. It has been incredibly helpful for refining my writing skills. We observed that the expression in the previous version was insufficiently standardized, and the ambiguous subjects has compromised the rigor of the data description.
4.During reviewing, I have found a similar paper entitled "Effect of Partial Substitution of Flour with Mealworm (Tenebrio molitor L.) Powder on Dough and Biscuit Properties by Xinyuan Xie, Zhihe Yuan, Kai Fu, Jianhui An, and Lingli Deng * July 2022 Foods 11(14):2156. DOI: 10.3390/foods11142156 has been published. How is your previous paper different from the current manuscript?
Response: Thank you for your suggestion. Firstly, the most notable difference lies in the use of distinct functional ingredients. Xie et al. added mealworm powder (with a protein content of 43.5%), whereas this study employed mealworm protein powder, which boasts a significantly higher protein content of up to 65%. As noted in lines 66–71, mealworm powder exhibits certain limitations in practical application and mealworm protein powder effectively addresses by highlighting its inherent advantages. Secondly, to systematically explore the feasibility of applying mealworm protein powder to biscuit and cater to the diverse demands of consumers, this study extended its research scope to two categories: soda crackers and cookies. Consequently, the research workload was approximately double that of Xie et al.’s study, and the expanded sample range renders the results more robust and convincing. Thirdly, by integrating relevant literature and the characteristics of biscuit, this study optimized the determination method for product texture. Specifically, a 3-point bending probe (TA/3PB) was selected in place of a P36R probe, enabling a more scientific reflection of texture changes in biscuits. Furthermore, Pearson correlation analysis and principal component analysis (PCA) were employed to synthesize the overall research findings. These statistical methods not only systematize the research results but also more comprehensively illustrate the effects of mealworm protein powder on dough properties and final product quality.
5.The English could be improved to more clearly express the Quality of Figures research.
Response: Thank you for your suggestion. We have checked the entire text and revised some inappropriate expressions. At the same time, we have engaged a professional language polishing agency to refine the content, with revisions made to the corresponding parts (modified parts are highlighted in blue).
- Figures and tables can be improved.
Response: Thank you for your suggestion. We have made adjustments to some of the Figures and tables to make them look clearer and more reasonable. Specifically, we uniformly presented the formulations in percentages in table 1. The Figures were adjusted and beautified to make it appear clearer and more understandable. Table 4 has been split into two separate tables, with soda crackers and cookies presented in each table respectively. The “p” in Figure 6 has been modified to be consistent with that in the text (p).

Reviewer 2 Report (Previous Reviewer 2)
Comments and Suggestions for Authors
The authors made a series of corrections and considerations to the work, which greatly improved its scientific character. However, there are some points that need to be revised, in my opinion, as cited below:
Line 35 - Grammatical correction of the word "diverses";
Table 1 - Always present the formulations in percentages, and clarify whether it is a total percentage or a flour base;
Line 265 - The authors tend to explain the behavior of the M curve but do not mention a possible cause for the behavior of the other curves; Please review;
Line 280 - The authors mention enriching a formulation with fiber to compare the results of this work with another previously published one; however, this comparison makes no sense, since the flour used has a very small amount of fiber in its composition, making the comparison impossible; The discussion should be based on its protein content;
Lines 359-362: Review the discussion, as only the value of "a" varied between the formulations;
Lines 369-374: Include in the discussion the difference in the composition of the two evaluated products, since the cookies have a higher fat and sugar content compared to the cracker;
Lines 424-426: The sentence presents an obvious fact rather than an explanatory one, since the products have very different chemical compositions;
Conclusion: The authors cannot compare the results between the two products due to the higher or lower addition of mealworm flour, since the products have very different chemical compositions...in fact, since they are completely different products.
Please review this topic in the discussion of results and especially in the conclusion.
Author Response
Comments:
1.Line 35 - Grammatical correction of the word "diverses";
Response: Thank you for your suggestion. This was indeed an oversight on our part, and we have made the necessary revisions.
2.Table 1 - Always present the formulations in percentages, and clarify whether it is a total percentage or a flour base;
Response: Thank you for your suggestion. We have presented the formulations in percentages in Table 1 and indicated "based on composite flour weight, 100%" in the table title. To avoid confusion, we have specified in lines 138–139 that the mass of the composite powder is 150 g.
3.Line 265 - The authors tend to explain the behavior of the M curve but do not mention a possible cause for the behavior of the other curves; Please review;
Response: Thank you for your suggestion. As the test time extends, other curves showed a relatively stable downward trend in the later stage. It is indicated that the starch-gluten network structure of the dough became unstable and began to decompose with the extension of mixing time. This led to a gradual reduction in the overall strength of the dough and a decline in its stability. We supplemented it in lines 267-269 and added corresponding reference.
4.Line 280 - The authors mention enriching a formulation with fiber to compare the results of this work with another previously published one; however, this comparison makes no sense, since the flour used has a very small amount of fiber in its composition, making the comparison impossible; The discussion should be based on its protein content;
Response: Thank you for your suggestion. Based on your suggestion, we had a careful discussion and decided to delete the description about fibers reducing water absorption rate. And the reasons for the changes in water absorption rate were enriched from the perspective of protein content. We have made the modifications in lines274-289。
5.Lines 359-362: Review the discussion, as only the value of "a" varied between the formulations;
Response: Thank you for your suggestion. After checking the manuscript, we found that it is indeed inappropriate to confuse the a* value and b* value of cookies. For this, we have revised the description of the changing trends of a* and b* and supplemented “Notably, minimal variation in the b* was recorded in cookies when the substitution level increased from 5% to 20%, suggesting that the concentration of mealworm protein powder had no substantial impact on b* in cookie formulations. However, a* also showed no significant change at a higher substitution level (> 5%), which is quite different from the trend of soda crackers. This might be because the high butter content in cookies can neutralize the color effect brought by different concentrations of mealworm protein powder.” in lines 367-374.
6.Lines 369-374: Include in the discussion the difference in the composition of the two evaluated products, since the cookies have a higher fat and sugar content compared to the cracker;
Response: Thank you for your suggestion. By consulting relevant materials, we found that the high fat and sugar content does indeed affect the stability of the gluten network and the gas retention of the product, thereby influencing the baking expansion ratio of cookies. We supplemented it in lines381-383 and added corresponding reference to support it.
7.Lines 424-426: The sentence presents an obvious fact rather than an explanatory one, since the products have very different chemical compositions;
Response: Thank you for your suggestion. After thinking and discussion, we believed that it is indeed inappropriate to overly focus on the fact that the two types of biscuits use the same composite powder. There are significant differences in the chemical composition of the two products due to the difference in auxiliary materials. For this, we have modified the original content in lines442-446 to “This is attributed to the fact that the two types of biscuits are added with different auxiliary materials, resulting in significant differences in chemical composition, which leads to certain differences in the correlation between dough characteristics and product quality.”
8.Conclusion: The authors cannot compare the results between the two products due to the higher or lower addition of mealworm flour, since the products have very different chemical compositions...in fact, since they are completely different products. Please review this topic in the discussion of results and especially in the conclusion.
Response: Thank you for your suggestion. We have checked the entire text and revised the inappropriate descriptions to better reflect the impact of mealworm protein powder itself on the product rather than focusing on comparing the quality differences between the two products. We have modified the content of "lines 526-530" to “Overall, the cookies have a relatively low hardness and show a low dependence on mealworm protein powder. These observations were further supported by multivariate statistical analysis. A higher sensory score indicated that the cookies better met the needs of consumers than soda crackers with mealworm protein powder did.” We have modified the original content in lines442-446 to “This is attributed to the fact that the two types of biscuits are added with different auxiliary materials, resulting in significant differences in chemical composition, which leads to certain differences in the correlation between dough characteristics and product quality.”

Round 2
Reviewer 1 Report (Previous Reviewer 1)
Comments and Suggestions for Authors
Dear Authors,
Thank you for improving the manuscript as per the suggestions.
Author Response
Thank you for your hard work.
This manuscript is a resubmission of an earlier submission. The following is a list of the peer review reports and author responses from that submission.
Round 1
Reviewer 1 Report
Comments and Suggestions for Authors
Dear Authors,
Thank you for submitting the manuscript entitled "Sustainable Protein Source Formulated Biscuit: Impact of Tenebrio molitor Protein Powder on the pasting, farinograph, and entensograph properties of wheat flour and Biscuit Quality" for the MDPI Foods Journal. Please check the major shortcomings of your manuscript.
What do you mean by low-gluten wheat flour? It would be helpful to specify the gluten content range of the flour used for biscuit preparation.
2.8 Analysis of the texture profile of bread. Check the title. You have prepared biscuits or cookies, not bread.
Line no. 264, The water absorption of the dough significantly decreased from 63% to 55% after replacing whole wheat flour with spinach flour containing higher dietary fiber. Are the findings of Waseem et al. [36] or any other authors? Please check and write the reference. As far as I know, when the dietary fiber is high, water absorption is also high. Dietary fiber, especially soluble fiber, tends to absorb water and form interactions with water molecules. Similar is true for protein also. Proteins also form bonds with water molecules. Please check the statement given by you from line numbers 257-265. It would be better if you are familiar with the functional and chemical characteristics of the Tenebrio molitor Protein Powder before adding it to the low-gluten wheat flour. This is because the findings or results you obtain after adding it to the flour are contradictory.
The degree of softening of flour is decreased after adding the Tenebrio molitor Protein Powder, but at 15% it is increased and then decreased. It showed that the mixture was not mixed homogenously or adequately. Please check the results.
Please explain Figure 3 in the results; only adding the figure is not sufficient. Write your findings and discuss the results with suitable references.
Please check Table 4 for the calculation of the stretch ratio. Highlighted in the manuscript, also. Table 4 and Figure 4 represent similar findings, so only the figure or the table can be kept.
This phenomenon was attributed to the formation of small ice crystals during the freezing stage, which caused volume expansion in cookie preparation. Subsequently, the ice crystals melted, leading to a final reduction in product volume during high-temperature baking. Please add a recent reference to support this finding.
The hardness of biscuits should increase due to the formation of a strong gluten network after the addition of protein. Please check your results, which are contradictory to the basic research. You have quoted in reference No. This discrepancy has been attributed to compromised gluten network formation upon insect powder addition, which leads to elevated dough density and promotes the tight combination of proteins and starches through hydrogen bonds. This also clarifies that after the addition of protein, a tougher network is formed by the combination of protein and starches through hydrogen bonds. It also suggests that the water absorption improved due to the bonding of water and protein. Please check the relevant references.
The Maillard reaction gave the cookies a more stable cross-linked structure. The Maillard reaction will not provide a stable cross-linked structure, which is due to a stronger gluten network. The Maillard reaction is the non-enzymatic reaction of an amino group with a reducing group (often a reducing sugar), leading to the formation of compounds which ultimately polymerise to form brown pigments. Please check.
However, cookies are characterized by low moisture content and high lipid content. Have you measured the moisture and fat content of cookies. If yes, then add the results into the manuscript.
Line no. 444, Based on the analysis of research findings, it can be assumed that Tenebrio monitor protein powder holds promise as an alternative protein source for biscuit production, given that its protein content is 56.5% higher per 100 g compared to low gluten wheat flour. You have not shown the results of protein content, so where did this statement come from?
Line no. 449, Furthermore, the incorporation of meal-worm protein powder decreases the water absorption and extensibility of dough, while simultaneously increasing the farinograph quality number and stretch ratio, which collectively indicate enhanced dough strength. Here, you are suggesting that dough strength was increased after the addition of mealworm protein, which shows that the gluten network has improved; therefore, only the dough strength can be improved. Please check the statement.
Sensory and proximate composition is missing.
Comments on the Quality of English Language
English can be improved.
Reviewer 2 Report
Comments and Suggestions for Authors
The article presents a study on the characterization of a wheat flour enriched with insect protein and its application in biscuit formulations. It is a very current work that meets the demand of the consumer market, which is eager for healthier, nutritionally enriched products, using an inexpensive and often underutilized protein source. However, this work has some flaws that need to be addressed for successful publication, as stated below:
1 - The title contains a grammatical error, is too long, and unattractive;
2 - Abstract: Rephrase lines 10-11;
3 - Lines 81-82: "Low gluten wheat flour is commonly used as the raw material for biscuit production due to its relatively low content of gluten protein." is a redundant sentence...rephrase;
4 - Item 2.6: Clarify the flour blend used to make the cookies (M0, M5, M10, M15, or M20?), as well as the reason and form it was selected.
5 - Figure 1: Unnecessary
6 - Line 369 "yellow Tenebrio molitor protein powder": Why yellow? This adjective had not been used until now... If it is a selling characteristic of the flour, it needs to be described in the materials and methods.
7 - Line 423: Verify;
8 - Lines 462-463: No sensory evaluation was performed on the products produced; no conclusions can be drawn about their smell.
9 - Could the authors clarify why they did not perform a sensory analysis? Volume of the products produced? Proximate composition? Shelf life? SEM? Such analyses would assist and complement the discussion of the data, giving more substance to the article.
After corrections, the work will present scientific merit for publication, but it is too outdated in terms of analysis to complement the discussion.
Reviewer 3 Report
Comments and Suggestions for Authors
Dear Authors,
Overall, the manuscript addresses a topic of current interest — the use of Tenebrio molitor protein in biscuit recipes — and has the potential to contribute to the field of food sustainability. However, the version presented has considerable weaknesses that affect its acceptance.
The title is too long and detailed. A more concise version that focuses on the main objectives of the study is recommended.
The summary is well structured, but there is a formatting error: the species name Tenebrio molitor is not fully italicized at the first mention. The taxonomic authorship of the species should be indicated: Tenebrio molitor (Linnaeus, 1758). After the first mention of the scientific name, it is recommended to use the common name (e.g. "mealworm" or "edible insect") or the abbreviation T. molitor to facilitate readability. The keywords should not repeat the terms from the title. The aim is to increase the visibility of the article and indexing, so it is recommended to use complementary or broader terms.
The introduction is excessively long. The first paragraph could be significantly shortened, as could the section dealing with the use of insects in food. Many examples of previous studies with insect meal are given, including quantitative results (e.g. increase in protein content), which would belong in the discussion rather than the introduction. The final paragraph details the objectives and mixes methodological considerations, which is inappropriate. It is suggested that it be reworded so that the objectives are stated briefly and clearly.
The description of the methodology is generally appropriate but lacks some important information; for example, the units of measurement for several parameters are missing. In section 2.7, entitled "Physiochemical analysis"," only the analyses of colour and expansion ratio — both physical characteristics — are described. The title should be corrected or supplemented by actual chemical analyses. No nutrient analyses were carried out. This absence is critical as the main argument of the article is the nutritional value of the flour substitute. Without analytical data on the nutrient composition (protein, fat, carbohydrates, fibre, etc.) it is impossible to assess the actual effect of the flour substitute. The absence of these analyses undermines the aim of the work and prevents a serious evaluation of its scientific contribution.
Final recommendation: Reject
Despite the interesting topic and some merits in the technical execution, the lack of nutritional data combined with structural problems and a thorough scientific discussion make this manuscript unsuitable for publication in the journal Foods in its current form.